# Effects of Different Extraction Methods on the Gelatinization and Retrogradation Properties of Highland Barley Starch

**DOI:** 10.3390/molecules27196524

**Published:** 2022-10-02

**Authors:** Mengzi Nie, Chunhong Piao, Jiaxin Li, Yue He, Huihan Xi, Zhiying Chen, Lili Wang, Liya Liu, Yatao Huang, Fengzhong Wang, Litao Tong

**Affiliations:** 1College of Food Science and Engineering, Jilin Agricultural University, Changchun 130118, China; 2Key Laboratory of Agro-Products Processing Ministry of Agriculture, Institute of Food Science and Technology, Chinese Academy of Agricultural Sciences, Beijing 100193, China

**Keywords:** highland barley starch, extraction method, gelatinization, retrogradation

## Abstract

The purpose of this study was to compare the gelatinization and retrogradation properties of highland barley starch (HBS) using different extraction methods. We obtained HBS by three methods, including alkali extraction (A-HBS), ultrasound extraction (U-HBS) and enzyme extraction (E-HBS). An investigation was carried out using a rapid viscosity analyzer (RVA), texture profile analysis (TPA), differential scanning calorimetry (DSC), X-ray diffraction (XRD) and Fourier-transform infrared spectrometry (FTIR). It is shown that the different extraction methods did not change the crystalline type of HBS. E-HBS had the lowest damaged starch content and highest relative crystallinity value (*p* < 0.05). Meanwhile, A-HBS had the highest peak viscosity, indicating the best water absorption (*p* < 0.05). Moreover, E-HBS had not only higher G′ and G″ values, but also the highest gel hardness value, reflecting its strong gel structure (*p* < 0.05). These results confirmed that E-HBS provided better pasting stability and rheological properties, while U-HBS provides benefits of reducing starch retrogradation.

## 1. Introduction

Highland barley (HB) is a high-quality raw material for the development of functional foods, among which the most abundant component is starch, accounting for around 65% by dry weight of grains [1]. The regular intake of resistant starch (RS) is known to protect against certain diseases and promote colon health [2]. Relevant studies have shown that the content of RS in highland barley starch could amount to 2.27–11.23%, and the content of slow-digestible starch (SDS) amounts to1.54–40.58% [3,4,5]. Good processing methods can increase the content of RS and SDS and then reduce the glycemic index, which is beneficial to the preparation of highland barley slow-digestible starch. For the extraction of starch, there are different extraction methods for starch separation from the grain, and the effect of different methods on starch are slightly different. The most commonly used method is alkali extraction (chemical extraction). Pires et al. found that starch obtained by alkali extraction gave higher yields than those from water extraction, but was harmful to the environment [6]. Moreover, alkali extraction has a relatively greater influence on the structure and properties of HBS, which is presumed to be detrimental to the maintenance of starch resistance. Among physical extraction techniques, ultrasound extraction is widely used thanks to its efficiency, chemical-free process and environmental friendliness [7,8]. Ultrasound could act on the amorphous regions of starch granules, changing the relative crystallinity (RC) and weakening the connections between starch molecules by breaking hydrogen bonds and double-helix structures, which in particular also led to a smaller average starch particle size [9]. In addition, some researchers have tried using enzymatic methods to release the starch granules. Buksa found that the yields of rye starch extracted by enzymatic methods (xylanase and protease) were much higher than those extracted by aqueous methods [10]. Ozturk et al. discovered the cellulase, xylanase and protease could cooperatively disrupt and loosen the network around the protein matrix or non-starch polysaccharides by micrographs, resulting in higher purity [11]. However, the implications of the different extraction methods on the physicochemical properties of HBS need to be further clarified. 

As is well known, the granular structure of starch is correlated with its physicochemical properties. The pasting behavior of starch is central to many starch-based food matrices and is generally characterized by changes in viscosity during the process of heating, holding and cooling [12]. The degree of gelatinization changes with different extraction methods of starch, which is related to the stability of the starch. Once the initial gelatinization temperature is reached, the granular and crystalline structure begins to break down, and the amylose gradually leaches out. The properties of starch paste were significantly different under different degrees of gelatinizing [13]. For the effect of starch extraction methods on the properties of starch paste, it was found the peak viscosity of Chinese yam starch obtained by aqueous extraction (SBS) was lower than that of those obtained by enzymatic and alkali extraction, which indicated that the water holding capacity of starch extracted by SBS was relatively poor [14]. After gelatinization, the differences in the structural properties of starch make the degree of retrogradation different, and then affect the properties of starch gel after retrogradation. In retrogradation, the starch chains recombine and form a double-helix structure during the cooling phase, which is then packed into crystals [15]. For now, there is little research about the effect of different extraction methods on the gelatinization and retrogradation of starch. The related research might be conductive for the development of grain products, such as bread, noodles and so on, especially for HBS. Therefore, the effects of different extraction methods of starch on the physicochemical properties and paste properties of HBS were explored to further improve the quality of cereal food: 1. to investigate the effect of different extraction methods on the gelatinization characteristics of HBS; 2. to clarify the effect of different extraction methods on the retrogradation characteristics of HBS; and 3. to analyze the influence of HBS structure changes on the gelatinization and retrogradation characteristics. 

## 2. Results and Discussion

### 2.1. Morphology and Chemical Compositions of HBS

The scanning electron micrographs of HBS obtained from three extraction methods are shown in Figure 1, which shows different granule morphologies. HBS consisted of large (most-part) and small (small-part) granules, similar to a previous report [3]. For all HBS samples, the particle size of the most-part granules were concentrated at 19.89–22.60 μm, and the size of the small-part granules were concentrated at 2.58–3.33 μm, as shown in Figure 2. This is consistent with previous reports that HBS granules’ diameter are generally 2–30 μm [16]. The surface of A-HBS (HBS extracted by alkali method) granules and the edge of U-HBS (HBS extracted by ultrasonic method) were relatively smooth, which indicated that the granular morphology was less damaged. In contrast, the E-HBS (HBS extracted by enzymatic method) granules were rough, with some fine fragments in morphology, which might be due to incomplete hydrolysis of partially insoluble dietary fiber (IDF) in the presence of enzymes. The hydrolysis of dietary fiber involved a synergistic attack of multiple enzymes, indicating IDF was hydrolyzed to form numerous cavities or spaces and that a dense network structure also hindered the enzymatic hydrolysis [17,18]. In addition to this, it was found the granular size distribution of A-HBS (D10, D50 and D90) was smaller than other granules (Table 1). The different granular morphology and size distribution of starch granules reflects different physicochemical properties, such as gelatinization and retrogradation behavior, swelling power and so on. [19].

As shown in Table 1, the purity of HBS obtained from all three extraction methods was larger than 90%, and there was little difference among these three starch samples. It was also found that the extracted method had little effect on amylose content. It is worth noting that the content of damaged starch in U-HBS was significantly higher than that of the other extraction methods, which meant that ultrasonic waves caused some damage to the starch granules. The other experimental values were quite consistent with those of Pina et al., who reported the following content ranges for crude protein, crude lipid, ash and total dietary fiber in HBS: 0.18–0.23%, 0.18–0.26%, 0.20–0.35% and 2.96–3.14%, respectively [20].

### 2.2. Pasting Properties 

Gelatinization is a vital property for starch-based food processing. The pasting curves for HBS were different (Figure 3) and the related parameters are listed in Table 2. The peak viscosity (PV) and final viscosity (FV) of A-HBS were significantly higher than HBS obtained from other extraction methods, which indicated the A-HBS had the best water absorption capacity [6]. According to the particle size distribution, the high water absorption capacity of A-HBS might be related to the larger specific surface area per unit weight of the relatively small granules [19]. Meanwhile, A-HBS had the lowest pasting temperature (PT), implying poor thermal stability [21,22]. The gelatinization curves of U-HBS and E-HBS were relatively gentle with smaller breakdowns (BV), indicating the starch paste had poor shear resistance. The value of BV was the most sensitive index of pasting properties, which represented the degree of starch granule breakage [23]. A higher BV value implied that more starch granules were broken during the heating process, and the internal starch molecules were released [24]. In contrast to A-HBS, U-HBS and E-HBS have a larger average particle diameter (Table 1). This means that water molecules could not easily enter inside the starch granules, leading to a lower swelling force and better stability during heating [25]. Meanwhile, it also suggested that U-HBS and E-HBS had better gelatinization stability and the starch paste, which was presumed to be due to the hydroxyl groups in HBS, combined more closely with the weak water and strengthened the double-helix structure of the starch, and thus the microcrystalline structure of the amylopectin was strengthened [23]. Moreover, U-HBS presented the lowest PV and FV, indicating an application for food products with restricted swelling, such as noodles. 

### 2.3. Gel Properties

As displayed in Table 2, the textural parameters of the HBS gels from different extraction methods were stored at 4 °C for 1 and 7 days. Starch retrogradation occurred during the cooling and storage process, through recrystallization of amylose and amylopectin recrystallization [26]. On the other hand, it can be seen directly through the pictures in Figure 4C that the gel of U-HBS-1d flowed downwards, verifying the results of hardness. The gel hardness of E-HBS was significantly higher than that of A-HBS and U-HBS, since E-HBS had the lowest damaged starch content and larger average particle diameter (Table 1), both implying it had better particle integrity, resulting in incomplete reaction of water with the starch granules [27]. The hardness of all HBS gels increased in varying degrees after storage for 7 days. According to relevant studies, there was a close correspondence between the hardness of starch gels and retrogradation, and the degree of retrogradation during storage was clearly identified by measuring the hardness of HBS gels [28]. Both A-HBS and U-HBS had a lower degree of retrogradation, allowing them to be used as raw material for rice noodles. The cohesiveness was an evaluation indicator of the starch gel maintaining its structural integrity after the first occlusion and reflected the internal gel bonding strength [28]. There was no significant difference in the cohesiveness and springiness of prepared HBS gels.

### 2.4. Rheological Properties of HBS

To further evaluate the gel systems of different HBS, the rheological properties were characterized. As the angular frequency increased, the storage modulus (G′) and the loss modulus (G″) both tended to increase gradually (Figure 4 A,B). Furthermore, the G′ values were higher than the G″ values, indicating that all the starch-containing systems had gelled and behaved as viscoelastic solids [29]. U-HBS and E-HBS showed higher G′ and G″ values than A-HBS, which also reflected the gel strength of HBS. In addition, it could also be found that the higher the values of PV and SV, the lower the values of G′ and G″, which corresponded to the results of pasting properties. The loss tangent (tan δ) values were also an essential rheological parameter, which were evaluated using viscoelastic properties and defined as the ratio of the G″ to the G′ [30]. The tan δ values of HBS samples were far less than 1, which exhibited good elastic behavior. However, the tan δ of starch remarkably exceeded 1, indicating that the elasticity junction zones of the starch gel were destroyed [31]. It is noteworthy that the tan δ value of the A-HBS gels was around 0.15, indicating that the A-HBS pastes had a weaker gel structure. There was also a study reporting that starch pastes with weak gel structures preferred to form more solid doughs and were able to produce starch noodles with excellent dripping properties [31].

### 2.5. Thermal Properties 

To explore the thermal properties of starch, thermograms of all HBS samples treated were revealed in Figure 5. The DSC characteristic values of HBS by different extraction methods are shown in Table 3, and the onset (To), peak (Tp) and conclusion (Tc) temperatures and gelatinization enthalpy (ΔH) of the HBS varied significantly. It was found that the gelatinization temperature (To, Tp and Tc) of E-HBS was highest and that of A-HBS was the lowest, which meant the E-HBS had the best thermal stability and A-HBS the worst. Moreover, the ΔH of E-HBS was highest, followed by A-HBS and U-HBS. This proved the results of the pasting properties: E-HBS had a more ordered molecular or stable crystalline structure. The degree of retrogradation of all HBS samples during the storage process could be expressed by the retrogradation enthalpy ΔH [32]. With longer storage times, the ΔH showed a regular rise, indicating an increased degree of retrogradation. It is worth noting that all U-HBS samples had the lowest ΔH and the difference in enthalpy between day 1 and day 7 decreased (Table 3). This implied that the extent and rate of re-crystallization slowed down towards a certain degree. The differences in ΔH noticed were due to the intrinsic differences in the starch molecular structure once the structure has been completely disrupted by gelatinization [33]. Additionally, regenerated starch had a lower gelatinization temperature compared to natural HBS due to its weaker crystallinity. In the following section, particle structure measurements were carried out to investigate the reasons for the differences in gelatinization and retrogradation properties.

### 2.6. X-ray Diffraction Pattern

XRD measurement was performed to further investigate the long-range ordered structure of HBS extracted by three methods; the results are shown in Figure 6A. The pattern of the all three HBS showed strong peaks at 2θ of 15.13°, 17.08°, 18.24° and 23.02°, which were characteristic diffraction peaks for the A-type crystalline structure [34]. A-type starch crystals were monoclinic and generally consisted of a double-helix structure formed by amylopectin. XRD patterns were also instrumental in determining the influence of different extraction methods on the crystallinity of starch granules. The relative crystallinity (RC) values were displayed in Figure 6A, which was basically consistent with the values of gelatinization enthalpy (Table 3); the order was as follows: E-HBS (26.37%) > A-HBS (24.78%) > U-HBS (21.04%). It was found that the RC value of U-HBS (21.04%) was lower than that of the two other methods, which was because ultrasonic wave treatment had a high damage intensity to the crystalline structure, and the internal space became loose [8]. The E-HBS had the highest ratio of double helix structure, indicating that the destruction of the crystalline structure of starch by biological extraction of starch is lower than that of physical and chemical methods, and that ultrasonic wave treatment could lead to the disintegration of long-range ordered crystallites, as Wang et al. reported that the RC directly reflected the long-range ordered crystal structure of HBS [4].

### 2.7. Short-Range Ordered Structure 

The FTIR spectra of HBS obtained by different extraction methods are shown in Figure 6B. There was a peak found in the range of 4000 cm^−1^ to 3300 cm^−1^, which was mainly attributed to the existence of -OH groups occurring normally in carbohydrates [35]. In addition, the -OH absorption peak in A-HBS (3492.96) shifted toward the high-frequency wave number, indicating that the hydrogen bonding of A-HBS was weaker [36], which also explained the phenomenon that A-HBS was more prone to absorb water and swell into a paste. The bands at 1047 cm^−1^ and 1022 cm^−1^ represent the crystalline ordered and amorphous regions of starch, respectively. To reveal short-range order, absorbance ratios were calculated at 1047/1022 cm^−1^ (*R*1047/1022) [37]. As seen from Table 1, there were marginal differences in HBS obtained by three extraction methods. It was hypothesized that the above findings might be due to differences in the distribution of the amylopectin branch chain length, which in turn affected the composition of the amorphous region [4]. The above observations revealed that the starch granules of E-HBS had a better molecular structure, resulting in a higher gel hardness (Table 2).

### 2.8. Principal Component Analysis (PCA)

The principal component analysis (PCA) was further applied in this study in order to understand the potential mechanism of pasting and retrogradation. Figure 7A shows the results of PCA on the pasting characteristics of all HBS samples. The two principal components, PC1 and PC2, were able to explain 69.89% and 22.01% of the variable, respectively, for the total score was 91.90. E-HBS was located in the positive quadrant of PC1 and PC2 and had good thermal stability due to its high gelatinization temperature and ΔH, as well as through viscosity (TV). In addition, the results of the loading plot in Figure 7A showed that the short-range ordered (R1042/1022) and RVA parameters were close to each other, indicating that these indicators were positively correlated in the pasting process of HBS. This was further combined with the analysis of different extraction methods with storage time and aging factors in Figure 7B. Overall, the contribution of PC1 and PC2 to the total variation was 90.50%, indicating that the planes of PC1 and PC2 largely reflected the main contribution of the response variables. The factors associated with starch retrogradation (mainly ΔH and hardness) were distributed to the right of PC1, in contrast to the location of A-HBS and U-HBS, suggesting that both had a role in retarding HBS retrogradation in this study.

## 3. Materials and Methods

### 3.1. Materials 

The HB kernels (cultivar: Kunlun 15) were purchased from Xinlvkang food Co., Ltd. (Xining, China). Cellulase (≥400 U/mg protein) was purchased from Yuanye Technology Co., Ltd. (Shanghai, China). Xylanase (≥6500 U/g protein) was purchased from Novozymes (China) Biotechnology Co., Ltd. (Tianjin, China). All other reagents were analytical grade.

### 3.2. Samples Preparation

HB kernels were milled and sieved through 100 mesh to obtain highland barley flour (HBF). Afterwards, HBS was extracted using three methods based on the conditions of the extraction. The obtained HBS samples were denoted as A-HBS, U-HBS and E-HBS, respectively.

#### 3.2.1. Alkali Extraction

According to Yang et al., with some modifications [5], HBF (100 g) was added to NaOH solution (0.125 M, 600 mL), which was soaked at room temperature for 5 h. After that, the mixture was centrifuged (3000× *g*, 20 min) to discard the supernatant and the top gray layer was removed. Then, the precipitates were washed with deionized water and ethanol three times to remove soluble impurities, respectively. The precipitate obtained after washing was collected as starch and dried at room temperature, followed by passing through a 100-mesh sieve. The yield was then calculated as the percentage of the HBF weight.

#### 3.2.2. Ultrasound Extraction

HBF (100 g) was added to 600 mL distilled water, which was treated by an ultrasonic device (KQ5200DV, Kunshan Ultrasonic instrument Co., Ltd., Kunshan China) working at a frequency of 20 kHz and input power of 200 W for 20 min. Then, the sample was shaken in a water bath at 50 °C for 5 h. In order to ensure that the purity of HBS was greater than 90%, it was sieved through 300-mesh (0.05 mm) before centrifugation and repeating the above steps.

#### 3.2.3. Enzyme Extraction

According to Ozturk et al., with some modifications [11], HBF (100 g) was added to 600 mL distilled water. Then, 200 U/g cellulase and xylanase were added, and the mixture was shaken in water bath at 50 °C for 8 h, repeating the above steps.

### 3.3. General Compositions and Particle Size

According to the standard methods described by AACC (2010), the contents of moisture, crude protein, crude fat, ash and total dietary fiber were determined. The contents of total starch, amylose and damaged starch were determined by an assay kit (K-SDAM, Megazyme International Ltd., Wicklow, Ireland). A laser particle size analyzer (MS3000, Malvern Instruments, Worcestershire, UK) was chosen to measure the particle size distribution of HBS samples.

### 3.4. Scanning Electron Microscopy (SEM)

The morphology of HBS granules was observed by a scanning electron microscope (SEM Hitachi S-570, Hitachi, Co., Ltd., Tokyo, Japan) under the voltage of 10 kV. All samples were sputter-coated with gold and then observed at magnifications of 500×. 

### 3.5. Pasting Property 

The pasting properties of HBS samples were analyzed by a rapid viscosity analyzer (RVA-TecMaster, Perten Instruments, Sydney, Australia), following the method of Qin et al. [38]. A 3.0 g HBS sample was dispersed (based on 14% moisture content) in 25 mL of distilled water and then the mixture was measured in an RVA tank. The suspension was stirred at 960 rpm for 10 s and then reduced to 160 rpm for 50 s. The temperature was initially maintained at 50 °C for 1 min, then increased to 95 °C at the rate of 9 °C/min and maintained at 95 °C for 2.7 min. Finally, it was cooled to 50 °C within 3.75 min, and remained there for 2 min. 

### 3.6. Texture Profile Analysis (TPA)

The retrograded starch samples stored at 4 °C for 1 and 7 d in Section 2.5 were collected to determine gel properties by the texture analyzer (TAXT plus, Stable Co., Godalming, UK) coupled with a P/0.5 cylinder probe [39]. The experiment parameters of the texture analyzer were set as follows: the pretest speed, test speed and the latter test speed were set at 1.0 mm/s, the strain was 50%, the trigger type was automatic and the trigger force was 5 g.

### 3.7. Rheological Characterization

The viscoelastic and flow behaviors were determined by a stress-controlled rheometer (Physica MCR 301, Anton Paar GmbH, Graz, Austria) equipped with parallel plate geometry and a 1 mm gap [40]. The samples were starch pastes prepared in Section 3.5; a fixed amount of sample was transferred to the rheometer plate and excess sample was carefully wiped away. Frequency scan experiments were carried out at 1% strain from 0.1 to 10 Hz in the linear viscoelastic region. 

### 3.8. Thermal Analysis

The thermal properties were measured by differential scanning calorimetry (DSC 8000, PerkinElmer, Norwalk, CT, USA) according to Chen et al., with some modifications [39]. 3 mg HBS was weighed, mixed with 10 μL distilled water in an aluminum pot and equilibrated at 4 °C overnight. The mixture was heated from 30 °C to 130 °C at a rate of 10 °C/min. A sealed empty pan was used as a reference. After testing, the samples were stored at 4 °C for 1 and 7 d, repeating the above assay measurements. 

### 3.9. X-ray Diffraction (XRD) Analysis

The crystalline structure of HBS samples extracted by different methods was measured by an X-ray diffractometer (XRD, Bruker D8 Advance, Salbuluken, Germany) equipped with a copper tube operating at 40 kV and 200 mA and producing Cu-Kα radiation at a wavelength of 0.1542 nm. The diffractograms were obtained by scanning from 5° to 45 ° (2θ) at room temperature at a rate of 10°/min and in steps of 0.02°. The crystalline peak and total area of the diffractogram were analyzed using MDI Jade 5.0 software (Materials Data, Inc., Livermore, CA, USA). The relative crystallinity was calculated as the percentage of the area of the crystalline region to the area of total diffraction, as represented in Equation (1):(1)$$\mathrm{RC}\;(\%)=\frac{T_{a}-A_{f_{a}}}{T_{a}}\times 100$$
where RC is the relative crystallinity, T_a_ is the total area and A_fa_ is the area of the amorphous fraction.

### 3.10. Fourier-Transform Infrared Spectrometry (FTIR) Analysis 

The HBS samples (2 mg) and the dried KBr powders (100 mg) were mixed and ground thoroughly. The mixture was then pressed into tablets and observed by Fourier-transform infrared spectroscopy (TENSOR 27, Borken, Germany). The blank background was completed by the KBr powder alone. The scanning conditions were 400–4000 cm**^−^**^1^ of wavelength, 4 cm**^−^**^1^ of resolution and 64 scans. 

### 3.11. Statistical Analysis 

Analysis of variance (ANOVA) was used to measure statistical differences by SPSS Version 16.0 software (IBM software, Chicago, IL, USA). Significant difference (*p* < 0.05) was determined using the Duncan procedure. PCA was performed to visualize gelatinization and retrogradation properties of three kinds of HBS samples, respectively. PCA was achieved using Origin 2018b software (Origin-Lab, Inc., Northampton, USA). All experiments were carried out at least in triplicate.

## 4. Conclusions

In conclusion, the gelatinization and retrogradation characteristics of HBS obtained by the different extraction methods were significantly different. It was attributed to the granular structure of the HBS. E-HBS had the highest relative crystallinity value, thus it exhibited better thermal stability, which increased the crispness of foods as cookies. On the other hand, A-HBS showed the weakest hydrogen bonding, which resulted in the highest viscosity and could be used as a food thickener. Moreover, U-HBS had the highest damaged starch content, as well as the lowest gelatinization enthalpy after 1 and 7 days of storage, respectively, providing anti-aging properties, which made it suitable for foods with a short shelf life such as bread or noodles. The findings above provide a theoretical basis for the development of HBS and highland barley food with different expected properties. 

## Figures and Tables

**Figure 1 molecules-27-06524-f001:**
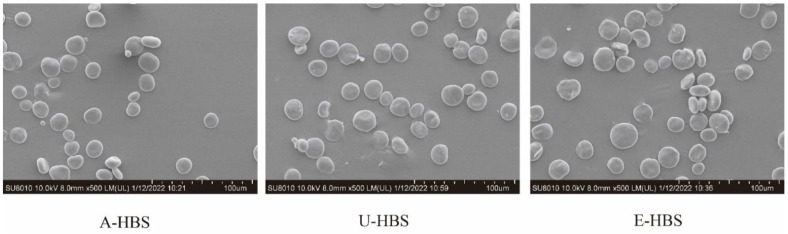
Scanning electron micrographs (SEM) (×500) of highland barley starch samples. A-HBS: HBS extracted by alkali method; U-HBS: HBS extracted by ultrasonic method; E-HBS: HBS extracted by enzymatic method.

**Figure 2 molecules-27-06524-f002:**
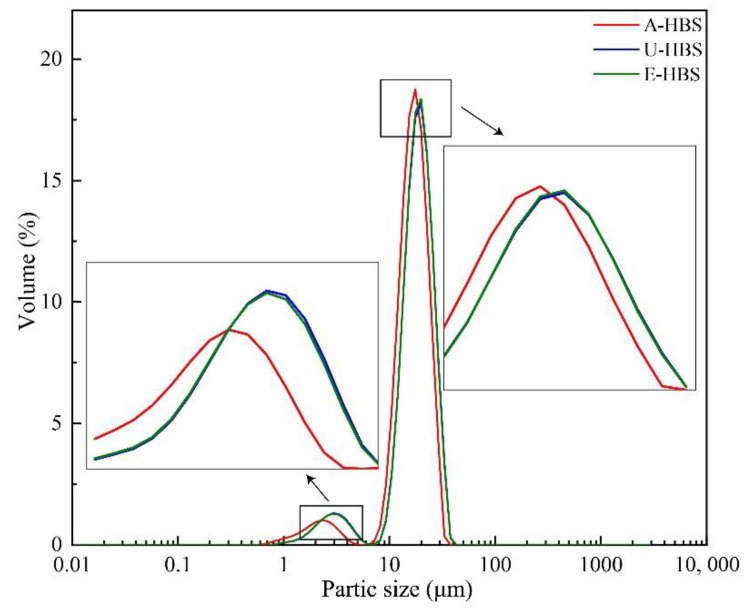
Particle size distribution of highland barley starch samples. A-HBS: HBS extracted by alkali method; U-HBS: HBS extracted by ultrasonic method; E-HBS: HBS extracted by enzymatic method.

**Figure 3 molecules-27-06524-f003:**
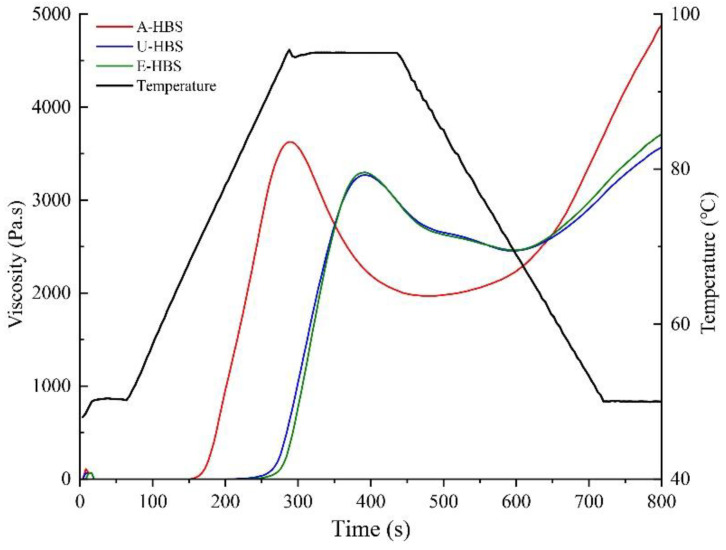
Pasting profiles of highland barley starch samples. A-HBS: HBS extracted by alkali method; U-HBS: HBS extracted by ultrasonic method; E-HBS: HBS extracted by enzymatic method.

**Figure 4 molecules-27-06524-f004:**
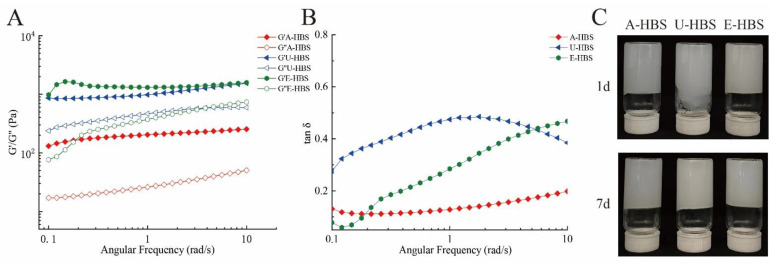
Rheological properties of highland barley starch samples: storage modulus (G′), loss modulus (G″) (**A**); tan δ (**B**) and pictures of HBS gels stored for 1 and 7 days (**C**). A-HBS: HBS extracted by alkali method; U-HBS: HBS extracted by ultrasonic method; E-HBS: HBS extracted by enzymatic method.

**Figure 5 molecules-27-06524-f005:**
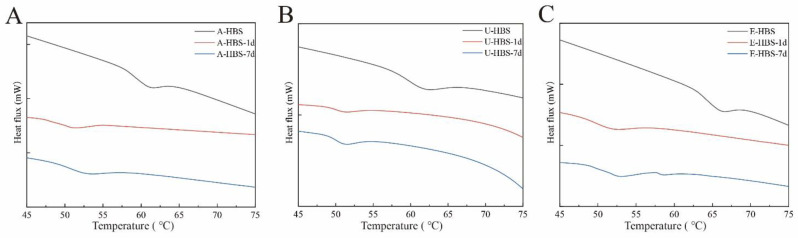
Differential scanning calorimetry (DSC) curves of A-HBS (**A**), U-HBS (**B**), E-HBS (**C**). A-HBS: HBS extracted by alkali method; U-HBS: HBS extracted by ultrasonic method; E-HBS: HBS extracted by enzymatic method.

**Figure 6 molecules-27-06524-f006:**
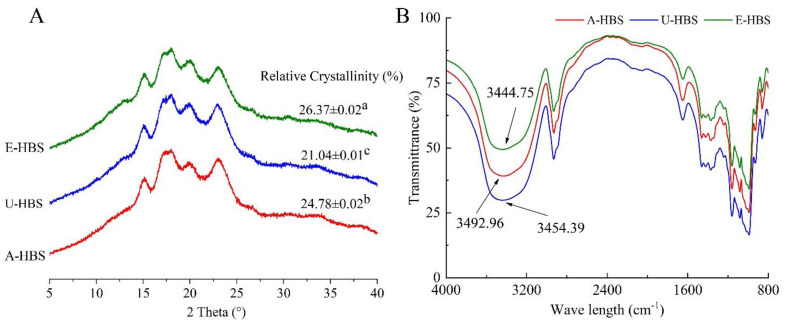
X-ray diffractograms (**A**), FTIR spectra (**B**) of highland barley starch samples. A-HBS: HBS extracted by alkali method; U-HBS: HBS extracted by ultrasonic method; E-HBS: HBS extracted by enzymatic method. Mean ± SD is calculated from triplicate measurements. Different lowercase letters represent significant difference (*p* < 0.05).

**Figure 7 molecules-27-06524-f007:**
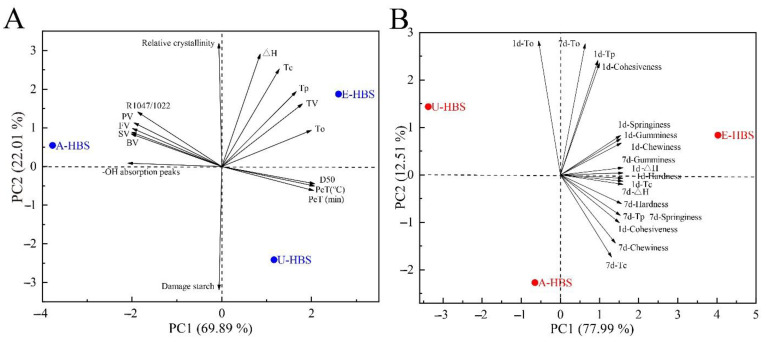
Principal component analysis*—*Correlation analysis based on different extraction methods with gelatinization (**A**) and retrogradation (**B**) properties. A-HBS: HBS extracted by alkali method; U-HBS: HBS extracted by ultrasonic method; E-HBS: HBS extracted by enzymatic method.

**Table 1 molecules-27-06524-t001:** Chemical composition of highland barley starch samples.

	A-HBS	U-HBS	E-HBS
Total starch (%)	92.33 ± 0.23 ^a^	92.15 ± 0.26 ^a^	90.68 ± 0.12 ^b^
Amylose starch (%)	23.37 ± 0.01 ^a^	23.71 ± 0.29 ^a^	23.76 ± 0.03 ^a^
Damaged starch (%)	1.66 ± 0.02 ^b^	2.57 ± 0.06 ^a^	1.20 ± 0.03 ^c^
Moisture (%)	8.46 ± 0.06 ^b^	8.65 ± 0.05 ^b^	9.14 ± 0.10 ^a^
Crude protein (%)	0.18 ± 0.01 ^a^	0.24 ± 0.02 ^a^	0.23 ± 0.02 ^a^
Crude lipid (%)	0.18 ± 0.04 ^a^	0.25 ± 0.00 ^a^	0.26 ± 0.02 ^a^
Ash (%)	0.35 ± 0.06 ^a^	0.25 ± 0.02 ^a^	0.20 ± 0.01 ^a^
Total dietary fiber (%)	3.04 ± 0.11 ^a^	3.14 ± 0.02 ^a^	2.96 ± 0.01 ^a^
Yield (%)	42.15 ± 0.33 ^a^	30.12 ± 0.27 ^c^	36.91 ± 0.18 ^b^
D10 (μm)	11.35 ± 0.13 ^b^	12.02 ± 0.03 ^a^	12.20 ± 0.13 ^a^
D50 (μm)	19.10 ± 0.12 ^b^	20.70 ± 0.00 ^a^	20.81 ± 0.10 ^a^
D90 (μm)	26.90 ± 0.13 ^b^	30.40 ± 0.03 ^a^	30.7 ± 0.00 ^a^
*R*1047/1022	1.15 ± 0.00 ^a^	1.11 ± 0.07 ^a^	1.12 ± 0.02 ^a^

A-HBS: HBS extracted by alkali method; U-HBS: HBS extracted by ultrasonic method; E-HBS: HBS extracted by enzymatic method. Mean ± SD is calculated from triplicate measurements. Different lowercase letters represent significant difference (*p* < 0.05).

**Table 2 molecules-27-06524-t002:** Pasting and textural properties of highland barley starch samples.

	A-HBS	U-HBS	E-HBS
Pasting properties			
PV (cP)	3597 ± 35 ^a^	3258 ± 16 ^b^	3311 ± 11 ^b^
TV (cP)	1935 ± 34 ^b^	2437 ± 43 ^a^	2489 ± 36 ^a^
FV (cP)	5180 ± 77 ^a^	3604 ± 59 ^c^	3773 ± 21 ^b^
BV (cP)	1662 ± 16 ^a^	820 ± 29 ^b^	822 ± 7 ^b^
SV (cP)	3244 ± 16 ^a^	1167 ± 24 ^c^	1284 ± 13 ^b^
PeT (min)	4.76 ± 0.14 ^b^	6.56 ± 0.11 ^a^	6.60 ± 0.10 ^a^
PeT (°C)	70.76 ± 0.01 ^c^	90.50 ± 0.04 ^b^	92.34 ± 0.10 ^a^
Textural properties			
	1 d		
Hardness (g)	210.28 ± 0.90 ^b^	183.47 ± 0.77 ^c^	258.58 ± 0.39 ^a^
Cohesiveness	0.43 ± 0.05a	0.44 ± 0.00 ^a^	0.45 ± 0.01 ^a^
Springiness (%)	91.03 ± 1.79 ^a^	90.61 ± 0.43 ^a^	94.74 ± 0.37 ^a^
Gumminess	92.58 ± 1.77 ^b^	89.43 ± 0.34 ^b^	114.28 ± 1.29 ^a^
Chewiness	84.32 ± 1.28 ^b^	80.76 ± 0.07 ^b^	103.34 ± 0.52 ^a^
	7 d		
Hardness (g)	283.77 ± 0.86 ^b^	219.75 ± 0.56 ^c^	348.30 ± 0.71 ^a^
Cohesiveness	0.58 ± 0.02 ^a^	0.54 ± 0.04 ^a^	0.60 ± 0.01 ^a^
Springiness (%)	94.33 ± 1.60 ^a^	92.53 ± 0.33 ^a^	95.83 ± 0.21 ^a^
Gumminess	127.45 ± 1.19 ^b^	112.85 ± 1.02 ^c^	153.91 ± 1.75 ^a^
Chewiness	134.54 ± 0.08 ^b^	112.05 ± 1.05 ^c^	139.74 ± 0.45 ^a^

A-HBS: HBS extracted by alkali method; U-HBS: HBS extracted by ultrasonic method; E-HBS: HBS extracted by enzymatic method; PV: peak viscosity; TV: trough viscosity; FV: final viscosity; BV: breakdown viscosity; SV: setback viscosity; PT (min): pasting time; PeT (°C): pasting temperature. Mean ± SD is calculated from triplicate measurements. Different lowercase letters represent significant difference (*p* < 0.05).

**Table 3 molecules-27-06524-t003:** Thermal properties of highland barley starch samples.

	To (°C)	Tp (°C)	Tc (°C)	ΔH (J/g)
A-HBS	57.03 ± 0.53 ^b^	61.15 ± 0.61 ^b^	65.46 ± 0.77 ^b^	5.69 ± 0.11 ^b^
U-HBS	59.76 ± 0.16 ^b^	61.99 ± 0.42 ^b^	65.08 ± 0.45 ^b^	5.32 ± 0.10 ^b^
E-HBS	62.41 ± 0.08 ^a^	65.63 ± 0.18 ^a^	69.30 ± 0.96 ^a^	6.48 ± 0.37 ^a^
A-HBS-1d	47.32 ± 0.74 ^b^	50.35 ± 0.53 ^a^	52.56 ± 1.93 ^a^	1.46 ± 0.18 ^b^
U-HBS-1d	48.10 ± 0.34 ^a^	50.92 ± 0.39 ^a^	51.88 ± 0.29 ^a^	1.13 ± 0.04 ^b^
E-HBS-1d	47.72 ± 0.64 ^b^	51.43 ± 0.78 ^a^	53.55 ± 1.63 ^a^	2.15 ± 0.12 ^a^
A-HBS-7d	47.62 ± 0.17 ^b^	52.35 ± 0.89 ^a^	54.97 ± 0.08 ^a^	2.27 ± 0.16 ^b^
U-HBS-7d	48.01 ± 0.28 ^a^	51.10 ± 0.56 ^b^	52.93 ± 0.32 ^b^	1.54 ± 0.18 ^c^
E-HBS-7d	48.15 ± 0.16 ^a^	53.15 ± 1.13 ^a^	55.12 ± 0.35 ^a^	3.25 ± 0.09 ^a^

A-HBS: HBS extracted by alkali method; U-HBS: HBS extracted by ultrasonic method; E-HBS: HBS extracted by enzymatic method; To: onset temperature; Tp: peak temperature; Tc: conclusion temperature; ΔH: enthalpy of gelatinization. Mean ± SD is calculated from triplicate measurements. Different lowercase letters represent significant difference (*p* < 0.05).

## Data Availability

Not applicable.

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
