# Peer review of "Effects of Different Extraction Methods on the Gelatinization and Retrogradation Properties of Highland Barley Starch"

_molecules, 2022, doi:10.3390/molecules27196524_

Round 1

Reviewer 1 Report

This study compared the gelatinization and retrogradation properties of HBS obtained with different methods. Several complementary techniques were used to measure the physicochemical properties of different HBSs. The study yielded adequate data for analysis but there are some major issues that need to be addressed before consideration for publication.

1. Line 28: give the full term when the word appears for the first time.

2. More discussions should be added besides simply presenting the results. In the pasting properties section, the authors should explain the potential reasons/mechanisms underlying the results obtained. Why U-HBS and E-HBS had better gelatinization stability and are not easy to retrograde? Does that have something to do with the particle size distribution and morphology too?

3. For Table 2, all the abbreviations need to be labeled in the table caption, i.e, PV, TV, SV, etc.

4. Line 143: "the storage modulus (G′) and the storage modulus (G′′)", so what is the difference between these two?

5. "The tan δ values" indicate what? What does it mean if the value exceeds 1? These need to be explained for clarity. 

6. Figure 4 has the wrong label - it should be figure 3. For Figure 3 A, which line represents which? The authors didn't specify G' and G'' between the two lines for each sample. 

7. Line 170: Here the authors indicate E-HBS had better particle integrity. However, in the previous particle size section, it was indicated that E-HBS had a rough surface and fragments. Doesn't that mean it presents lower structural or particle integrity? 

8. Line 179: From the results, the springiness values were not significantly different among the three. 

9. Cannot find Table 3. Is that included??

10. It would be better to include DSC thermograph figures to illustrate thermal properties.

11. Line 213-214: Were statistical analyses done for these crystallinity values?

12. Line 234: These bands did not appear in the figure. The authors need to increase the range of X-axis of Fig4B.

13. Line 299: Fix sentence.

14. How did the authors calculate crystallinity? This also needs to be included in the method. 

Reviewer 2 Report

General remarks: English language generally should be improved, some sentences are not written correctly (please see the attached file). The wrongly chosen word "regeneration" is used in many places in the manuscript. Please change it to the correct word "retrogradation” throughout the whole manuscript. Space should be added between the number and measurement unit throughout the whole manuscript, for example, 0.125 M.

Results and Discussion: Table 3 is mentioned several times in the manuscript; however, there is no table 3 in the manuscript. Please add it. Section 2.4. “Gel properties” should be moved after the pasting properties (2.2) and before the rheological properties.

Material and Methods: Lines 272-274: Please delete this sentence; it is unnecessary since it does not belong to this section. Please check the mesh of the sieves used for the sieving of starch samples. Please describe the methodology used to perform the principal component analysis (PCA) in the Statistical analysis section.

Tables and Figures: Table 1: There is no visible scale on micrographs. Please make them larger so that the starch granule morphology and scale can be easily seen. Table 2: The table caption should be placed above the table. Text in lines 137-139 should be added to the caption. Please define the abbreviations for pasting properties in the table caption. Table 3. It is completely missing from the manuscript, so please add it.

Conclusions: The final food products from highland barley starch should be mentioned, along with an explanation of which treatment is the best for which type of food, i.e. U-HBS is suitable for noodles (as stated in lines 132-134).

Round 2

Reviewer 1 Report

The authors have successfully addressed all the comments made previously, and the presentation of results has been significantly improved. Therefore, the manuscript in this form can be considered for publication.